# SELENOF Controls Proliferation and Cell Death in Breast-Derived Immortalized and Cancer Cells

**DOI:** 10.3390/cancers15143671

**Published:** 2023-07-19

**Authors:** Roudy C. Ekyalongo, Brenna Flowers, Tanu Sharma, Alexandra Zigrossi, An Zhang, Anaisa Quintanilla-Arteaga, Kanishka Singh, Irida Kastrati

**Affiliations:** Department of Cancer Biology, Loyola University Chicago, Maywood, IL 60153, USA

**Keywords:** breast cancer, selenoprotein F (SELENOF), tumor suppressor, acinar growth, proliferation, cell death

## Abstract

**Simple Summary:**

Selenoproteins are an elite family of proteins containing selenium in the form of the rare amino acid selenocysteine. SELENOF, formerly known as SEP15, is one of the selenoproteins that is highly sensitive to bioavailable selenium, which itself correlates with survival of breast cancer patients. SELENOF was reported to be lower or lost in aggressive breast tumors. Based on this evidence, SELENOF-driven functions and phenotypes were examined in normal breast epithelial and breast cancer cells to better understand the role of SELENOF in breast cancer. We found that loss of SELENOF confers features of oncogenic transformation while the opposite features are observed with its overexpression. These findings indicate that loss of SELENOF observed in breast tumors contributes to breast tumorigenesis.

**Abstract:**

SELENOF expression is significantly lower in aggressive breast tumors compared to normal tissue, indicating that its reduction or loss may drive breast tumorigenesis. Deletion of SELENOF in non-tumorigenic immortalized breast epithelial MCF-10A cells resulted in enhanced proliferation, both in adherent culture and matrix-assisted three-dimmensional (3D) growth. Modulation of SELENOF in vitro through deletion or overexpression corresponded to changes in the cell-cycle regulators p21 and p27, which is consistent with breast tumor expression data from the METABRIC patient database. Together, these findings indicate that SELENOF affects both proliferation and cell death in normal epithelial and breast cancer cells, largely through the regulation of p21 and p27. In glandular cancers like breast cancer, the filling of luminal space is one of the hallmarks of early tumorigenesis. Loss of SELENOF abrogated apoptosis and autophagy, which are required for the formation of hollow acini in MCF-10A cells in matrix-assisted 3D growth, resulting in luminal filling. Conversely, overexpression of SELENOF induced cell death via apoptosis and autophagy. In conclusion, these findings are consistent with the notion that SELENOF is a breast tumor suppressor, and its loss contributes to breast cancer etiology.

## 1. Introduction

SELENOF, previously known as SEP15, is a selenoprotein that contains selenium in the form of the amino acid selenocysteine [1]. Its role in carcinogenesis has been investigated due to its altered expression in tumors compared to corresponding normal tissue and the association of genetic variations in the *SELENOF* gene with cancer risk or outcome (reviewed in [2,3]). SELENOF has been implicated in breast cancer based on the observation that loss of heterozygosity at the *SELENOF* locus occurs in breast tumors [4]. The *SELENOF* gene is located on chromosome 1 at the 1p31 locus [1]. This region is commonly deleted in breast tumors, indicating that it may harbor putative tumor suppressors [5]. SELENOF is also highly responsive to selenium bioavailability, and a summary of the recent largest-to-date clinical and epidemiological studies indicates that selenium status correlates with breast cancer survival [2]. Based on these findings, the role of SELENOF in breast cancer was investigated [6].

Recent data supports a tumor suppressor function of SELENOF in breast cancer [6]. SELENOF mRNA was significantly lower in late-stage breast tumor samples compared to normal breast tissue, and lower SELENOF levels predicted poor patient outcomes from the disease [6]. Enhancing SELENOF expression in breast cancer cells attenuated aggressive disease phenotypes and elicited anti-cancer activity in vivo [6]. In breast and other examined tissues, except for the prostate, SELENOF is predominantly found in the endoplasmic reticulum (EnR) through binding to UDP-glucose:glycoprotein glucosyltransferase (UGGT), a folding sensor of the calnexin cycle [7,8]. The selenocysteine amino acid of SELENOF resides within a thioredoxin-like fold, indicating that SELENOF mediates redox functions [9]. Given SELENOF’s cellular localization within the EnR and its redox activity, it is postulated to play a putative role in disulfide-bond formation and protein redox quality control [10,11]. More recently, depletion of SELENOF in cultured human cells or in Selenof knockout animal models was shown to result in differential expression of metabolism-related genes and associated phenotypes [12,13,14,15,16,17]. In this study, SELENOF-driven functions and phenotypes were examined in normal breast epithelial and breast cancer cells to better understand the role of SELENOF in breast cancer and to identify potential new therapeutic strategies.

## 2. Materials and Methods

### 2.1. Reagents

Doxycycline (Dox), geneticin, puromycin, 5-ethynyl-2′-deoxyuridine (EdU), propidium idiodide (PI), Calcein AM, Hoechst 33342, ZVAD, chloroquine, and K03681were purchased from Sigma (St. Louis, MO, USA), Invitrogen (Waltham, MA, USA), ThermoFisher Scientific (Waltham, MA, USA), and Selleck Chemicals (Houston, TX, USA). Antibodies specific to SELENOF (ab124840), p21 (10355-1-AP), p27 (V9001-20UG), and β-Actin (A5441) were purchased from Abcam (Boston, MA, USA), Proteintech (Rosemont, IL, USA), Bioreagents (San Diego, CA, USA), and Sigma, respectively. siRNA for SELENOF (4392420, ID: s17999), p21 (#6456), and p27 Kip1 (#12324) were purchased from Thermo Fisher Scientific and Cell Signaling (Danvers, MA, USA).

### 2.2. Cell Lines, Culture Conditions and Treatments

MCF-10A, HCC70, MDA-MB-157, and MCF-7 cells were purchased from ATCC. The cells were maintained at 37 °C in a humidified atmosphere with 5% CO_2_, and the media were changed every 2–3 days. MCF-10A SELENOF knockout cells were generated using CRISPR-CAS9 genome editing. The single guide (sg) RNAs for SELENOF or non-targeting sgRNA controls were annealed and cloned into LentiCRISPRv2 (#52961, Addgene, Watertown, MA, USA). The lentivirus was produced by transfecting 1 µg of LentiCRISPR vector, 1 µg of pCMV-VSVg, and 1 µg of psPAX2 into 293T cells in a 60-mm dish using polyethylenimine. The viruses were harvested 48 h post-transfection and used to transduce MCF-10A cells. The medium was supplemented with 0.5 µg/mL of puromycin for 3 days post-transduction. Following selection, single cells were seeded in 96-well plates, and clones were expanded. SELENOF knockout efficiency was analyzed using Western blotting. MCF-10A or SELENOF KO clones were maintained in DMEM-F12 medium (Gibco, Waltham, MA, USA) supplemented with 10% fetal bovine serum (Gibco), 20 ng/mL epithelial growth factor, 0.5 mg/mL hydrocortisone, 100 ng/mL cholera toxin, and 10 µg/mL insulin. MCF-7 cells overexpressing SELENOF (MCF-7 SELENOF cells) were generated through stable transfection of MCF-7 cells with a Dox-inducible SELENOF expression construct, pRetroX-Tight-Pur-TetOn-Advanced, from Clontech, according to manufacturer’s instructions [6]. MCF-7 SELENOF cells were maintained in RPMI 1640 (Gibco) medium with phenol red supplemented with 10% fetal bovine serum, 1% non-essential amino acids, 2 mmol/L L-glutamine, 1% penicillin–streptomycin antibiotics, 6 ng/mL insulin, and selection antibiotics: 250 µg/mL geneticin and 1 µg/mL puromycin. To induce SELENOF expression, the cells were treated with 1 µg/mL Dox. HCC70 cells were maintained in phenol-red RPMI 1640 medium (ATCC, Manassas, VA, USA) supplemented with 10% fetal bovine serum, 1% non-essential amino acids, 1% L-glutamine, and 1% penicillin–streptomycin. The MDA-MB-157 cancer cells were maintained in phenol-red Leibovitz’s L-15 medium (ATCC) supplemented with 10% fetal bovine serum and 1% penicillin–streptomycin. Prior to the experiments (48–72 h), all the antibiotics in the media were withdrawn. Inhibitor concentrations for K03681, ZVAD, and CQ were based on the published literature [18,19,20]. All cell lines were tested for mycoplasma using the LookOut PCR detection kit from Sigma.

### 2.3. Cell Viability

The cells were seeded in 24-well plates, and medium was refreshed every 2–3 days. The cells were stained with 1% crystal violet in methanol and water (1:4) and solubilized in 1% sodium dodecyl sulfate. The absorbance was measured at 570 nm using a Polarstar Omega fluorescence plate reader (BMG Labtech, Ortenberg, Germany).

### 2.4. FACS Analysis

For 5-ethynyl-2′-deoxyuridine (EdU) incorporation, cells were stained according to the manufacturer’s instructions using an EdU Staining Proliferation Kit iFluor 488 (ab219801, Abcam). For cell cycle analysis, the cells were collected using trypsin and suspended in Hanks balanced salt solution buffer with 2% FBS, then washed twice with ice-cold PBS. The cells were then incubated with 100 μg/mL RNase and 100 μg/mL propidium iodide (PI) for 15 min and analyzed using FACS Canto (Beckman Coulter, Brea, CA, USA). Single cells were gated and analyzed to differentiate between G0/G1, S, G2 + M phases, and sub-G0/G1 using FCS Express 6 Plus (De Novo software, Los Angeles, CA, USA). Cell death was measured using annexin V and PI staining. The cells were washed with ice-cold PBS and suspended in 100 μL of staining solution containing 5 μL of annexin V-Alexa-488-conjugate. After 15 min of incubation at room temperature, 400 µL of fresh binding buffer was added per sample containing 1 μL of 20 μg/mL PI, then immediately analyzed using the FACS Canto instrument (Beckman Coulter).

### 2.5. RT-Quantitative PCR (QPCR)

Total RNA was isolated using TRIzol according to manufacturer’s instructions. RNA (0.5 µg) was reverse transcribed in a total volume of 10 μL using 200 U of M-MLV reverse transcriptase, 100 ng of random hexamer, 0.5 mM of deoxy-NTP, and 10 mM of DTT. The resulting cDNA was mixed with SYBR Green master mix, and the forward and reverse primers and amplifications were performed using a QuantStudio3 instrument (ThermoFisher Scientific) according to the manufacturer’s instructions. The fold-change was calculated using the ΔΔCt method, with β-actin serving as the internal control. All QPCR primers used were validated and are available upon request.

### 2.6. 3D Acinar Assay

24-well plates were coated with 200 μL of ice-cold Corning Matrigel basement membrane matrix, phenol red-free, and LDEV-free (356237) and maintained at 37 °C for 30 min. Single cell suspensions of 1.5 × 10^4^ cells were prepared in 5% ice-cold Matrigel and plated on the coated wells. A total of 150 μL of culture medium was added every 48 h during the 20-day growth period. Three-dimensional acinar structures larger than 100 µm in diameter were counted, and their size was quantified using an Olympus phase contrast microscope. For fluorescence microscopy imaging, the acini were stained for 20 min at room temperature in the dark with PI, Hoechst 33342, and Calcein AM dyes. Protein extraction from 3D acini was performed according to a published protocol [21].

### 2.7. Western Blot

Whole-cell extracts were prepared using the M-PER reagent (ThermoFisher Scientific). Proteins were separated using SDS-PAGE (Invitrogen), transferred to nitrocellulose membranes using an iBlot 2 instrument (Invitrogen), blocked for 1 h in TBS/T buffer containing 5% non-fat dry milk, and incubated with the appropriate primary antibody (diluted in 5% milk in TBS/T) overnight. The next day, the corresponding secondary antibody was applied for 1 h, and the signal was visualized using an iBright CL1000 Imaging System (Invitrogen) with the Pierce Supersignal West Pico chemiluminescent substrate (ThermoFisher Scientific).

### 2.8. Transmission Electron Microscopy (TEM) Imaging

MCF-7 SELENOF cells were washed with PBS, then immersed in PBS containing 2.5% paraformaldehyde and 2% glutaraldehyde for 1 h at room temperature. After washing with PBS and deionized water, the samples were fixed with deionized water containing 1% osmium tetroxide and 1.5% potassium ferricyanide for 1 h at 4 °C in the dark. The samples were then stained en bloc with 1% uranyl acetate, then dehydrated through incubation in an ascending series of alcohol concentrations (25, 50, 75, 95, 100%). The samples were incubated in epoxy resin comprised of a mixture of EMbed 812, nadic methyl anhydride, dodecenyl succinic anhydride, and 2,4,6-Tris(dimethylaminomethyl)phenol for 1 h at room temperature on a rotary mixer (Ted Pella Inc., Redding, CA, USA). After removing the epoxy resin, the embedding capsules containing epoxy resin were inverted, then placed onto relevant areas of the cancer cell monolayers. The epoxy resin was allowed to polymerize at 37 °C for 12 h, then at 60 °C for 48 h. Next, the samples were cooled to room temperature, and the embedding capsules containing the cancer cell monolayers were removed from the culture dish. Ultra-thin sections (70 nm) were cut using an ultramicrotome (EM UC7, Leica Microsystems, Wetzlar, Germany), mounted on formvar- and carbon-coated 200 mesh copper grids, then stained with filtered 1% uranyl acetate and Reynold’s lead citrate prior to imaging. The samples were imaged using a Philips CM 120 transmission electron microscope (TSS Microscopy, Hillsboro, OR, USA) equipped with a BioSprint 16-megapixel digital camera (Advanced Microscopy Techniques, Woburn, MA, USA). The images were acquired at magnifications of 10,000×.

### 2.9. Statistical Analysis

The data are presented as mean ± SEM from a minimum of three independent determinations. Statistical analysis consisted of 1- or 2-way ANOVA followed by Tukey posttest, or a *t*-test when applicable.

## 3. Results

Loss of SELENOF promotes the proliferation of MCF-10A cells in both adherent and extracellular matrix-supported three-dimensional culture.

The cellular impact of SELENOF loss in non-transformed immortalized human mammary epithelial MCF-10A cells [22], which have high endogenous levels of SELENOF (Figure 1A, first lane), was utilized as a well-established model to assess the functional consequences of aberrations found in cancer [23]. Several stable CRISPR-Cas9 SELENOF knockout (KO) lines were derived from single-cell clones, with all but one exhibiting no detectable SELENOF (Appendix A). Two SELENOF KO clones were selected and used throughout our studies (Figure 1A). Loss of SELENOF resulted in a significant increase in cell proliferation over six days (Figure 1B and Appendix A). Furthermore, the incorporation of 5-ethynyl-2′-deoxyuridine (EdU), a nucleoside analog, into newly synthesized DNA was significantly higher in SELENOF KO clones compared to MCF-10A wild-type (WT) control cells (Figure 1C), as was the expression of *Ki67* a proliferation marker (Figure 1D). Transient silencing of *SELENOF* using siRNA in HCC70 and MDA-MB-157 breast cancer cells resulted in a similar increase in *Ki67* measured using RT-QPCR (Appendix A). When MCF-10A cells were seeded in extracellular matrix (ECM)-rich 3-dimmesional (3D) culture, acini-like spheroids formed, characterized by a hollow lumen and recapitulating several aspects of mammary gland architecture in vivo [24,25]. Deletion of SELENOF resulted in the formation of more acini (Figure 2A,B) and larger acini compared to WT controls (Figure 2A,C) in varying ECM concentrations. In summary, SELENOF ablation increased the proliferation of normal non-tumorigenic and breast cancer cells in both standard adherent and 3D culture settings.

## 4. SELENOF Controls Proliferation and Cell Death via p21 and p27

The increased proliferation resulting from the loss of SELENOF prompted an examination of changes in cell cycle distribution and regulators of the cell cycle. SELENOF KO cells displayed a significant accumulation in the S-phase, consistent with their increased proliferation (Figure 3A). Furthermore, the loss of SELENOF led to reduced levels of the cyclin-dependent kinase inhibitor proteins, p21 and p27 (Figure 3B). A similar pattern of p21 and p27 expression levels was observed in 3D acini derived from SELENOF KO cells (Appendix A). The inhibitory p21 and p27 proteins block the activity of cyclin-dependent kinase 2 (CDK2) [26]. Treatment of SELENOF KO cells with the CDK2 inhibitor, K03861, resulted in a significant reduction in cell viability (Figure 3C) and enhanced cell death (Figure 3D and Appendix A) compared to isogenic WT control cells. In contrast, when SELENOF was overexpressed in MCF-7 breast cancer cells, which have low endogenous levels of SELENOF, using a doxycycline (Dox)-inducible system [6], there was a significant increase in p21 and p27 levels (Figure 3E). A non-specific effect of Dox was ruled out in control cells (Appendix A). Silencing either p21 or p27 (Appendix A) significantly rescued the previously reported cell death phenotype associated with SELENOF overexpression in these cells [6] (Figure 3F). Consistent with this observation, *SELENOF* mRNA was found to negatively correlate with *Ki67* and positively correlate with *p21* and *p27*, as determined using the METABRIC breast cancer patients’ database (Figure 4A). In METABRIC tumor samples, the top 400 genes co-expressed with SELENOF were identified, and pathway enrichment analysis was conducted. Pathways enriched in genes negatively correlating with SELENOF included ‘*Mitotic Cell Cycle*’ and ‘*DNA Replication*’ (Figure 4B, left panel), while pathways enriched in genes positively correlating with SELENOF included ‘*Regulation of Cell Cycle*’ (Figure 4B, right panel). Other top enriched pathways were ‘*Translation*’, ‘*Endoplasmic Reticulum to Golgi Transport*’, and ‘*Protein Folding*’ (Figure 4B), which are consistent with previous reports of SELENOF’s role in protein folding and redox quality control [11,17]. Together, these data indicate that the regulation of p21 and p27 contributes to SELENOF-driven effects on the cell cycle, proliferation, or cell death.

## 5. SELENOF Is Required for Lumen Formation in 3D Acini

MCF-10A epithelial cells form hollow acini when cultured on ECM in vitro [24,25]. After several rounds of cell division, the monolayer of cells in contact with ECM establishes apicobasal polarity. These cells withdraw from the cell cycle and initiate lumen formation [27]. Apoptosis was shown to play a major role in the formation and maintenance of the hollow glandular architecture [27]. In addition, TRAIL-mediated autophagy also contributes to lumen formation [28]. Fluorescence microscopy images of MCF-10A WT cells cultured in ECM demonstrated a glandular architecture consisting of live cells stained with calcein AM and dead cells stained with propidium iodide (PI) located in the center of the lumen (Figure 5A, first column). The addition of inhibitors of apoptosis and autophagy, ZVAD and chloroquine, respectively, reversed lumen clearing in MCF-10A WT cells (Figure 5A, second column), as expected. Upon deletion of SELENOF, the resulting acini failed to undergo lumen clearing (Figure 5A, third and fifth columns). The addition of apoptosis and autophagy inhibitors had no effect (Figure 5A, fourth and sixth columns). In adherent 2D culture, serum deprivation of cells induced significant cell death, typically via autophagy [29], as was evident in MCF-10A WT cells (Figure 5B, white bars). However, SELENOF KO cells were significantly less susceptible to this type of cell death (Figure 5B, grey bars). These findings indicate that SELENOF is required for lumen formation in 3D cultures and that loss of SELENOF confers resistance to apoptosis and autophagy in MCF-10A cells.

## 6. SELENOF Overexpression Induces Apoptosis and Autophagy

SELENOF overexpression in breast cancer cells reduced cell viability (Appendix A) and elicited anti-cancer activity in a xenograft model of breast cancer [6]. Upon induction of SELENOF overexpression in MCF-7 SELENOF cells using Dox, transmission electron microscopy was used to evaluate the ultrastructure of these cells. Swelling of mitochondria and rupture of mitochondrial outer membranes were noted in SELENOF overexpressing cells compared to control cells (Figure 6A, top panel). Furthermore, SELENOF overexpression generated a significant increase in fragmented nuclei, as quantified using Hoechst 33,342 staining (Figure 6B and Appendix A). These findings collectively indicate enhanced apoptosis with SELENOF overexpression. Phagosome formation was also noted in cells with SELENOF overexpression (Figure 6A, bottom panel and Appendix A), along with an increase in the conversion of LC3-II, a phagosome maker (Figure 6C). The addition of either an apoptosis inhibitor (ZVAD) or autophagy inhibitor (CQ) partially reversed cell death, whereas combining both inhibitors fully prevented the cell death induced by SELENOF overexpression (Figure 6D). Contribution from ferroptosis to the observed cell death was ruled out (Appendix A). These findings indicate that SELENOF overexpression induces cell death in breast cancer cells through both apoptosis and autophagy.

## 7. Discussion

New evidence on the role of SELENOF in breast cancer is presented here through the use of both loss-of-function and overexpression studies. The levels of the cyclin-dependent kinase inhibitor proteins, p21 and p27, were found to be reduced in SELENOF KO cells, consistent with data obtained from the mammary glands of Selenof knockout mouse [6]. A strong correlation between SELENOF and p21 or p27 expression was also observed in the METABRIC database of human breast tumor samples. Pathway analyses of genes co-expressed with SELENOF in these breast tumors implicate SELENOF in the regulation of cell cycle and replication, aligning with the in vitro observations. In contrast to the findings in normal mammary epithelium or breast cancer cells, where depletion or loss of SELENOF enhances proliferation, silencing SELENOF using siRNA in HeLa cells led to the inhibition of proliferation [10]. Intriguingly, SELENOF deficiency in HeLa cells affected the cell cycle by upregulating p21 and p27. These opposing consequences of SELENOF deficiency in cultured cells have been previously observed [17], although the reasons for the different tissue-specific outcomes remain unclear. In line with the SELENOF loss-induced reduction in p21 and p27, inhibition of CDK2 resulted in attenuated cell viability and increased cell death. These findings suggest that breast tumors with low/no SELENOF expression may be more therapeutically susceptible to CDK2 inhibitors. Currently, CDK2 is considered a potential target for therapy in breast cancer, and more selective CDK2 inhibitors are in preclinical development [30]. This newly identified therapeutic vulnerability requires further investigation in additional models in future studies.

Loss of SELENOF detrimentally affected the formation of lumen or hollow acini in MCF-10A cells, a process known as ductulogenesis [24,25]. In glandular cancers like breast cancer, tumor cells are displaced from their normal matrix niches in the early stages of tumorigenesis as they proliferate into the lumen of hollow glandular structures [31]. Filling of the luminal space is one of the hallmarks of early tumorigenesis. Seminal work by Brugge and colleagues demonstrated that perturbations in individual cell proliferation or apoptosis pathways alone are insufficient to induce luminal filling. However, oncogenic dysregulation of both proliferation and apoptosis (e.g., by ErbB2) is sufficient to drive this process [32]. The observation that SELENOF is required for lumen formation in 3D cultures indicates its role in controlling both proliferation and cell death. Intriguingly, matrix-detached cells display dysregulation of metabolic activities and reactive oxygen species (ROS) [31], suggesting putative functions of SELENOF in redox and metabolism that may be relevant to breast tumorigenesis [17].

## 8. Conclusions

SELENOF levels are significantly lower in late-stage breast tumors, and lower SELENOF expression predicts poor patient outcomes [6]. The impact of SELENOF loss was investigated using non-transformed immortalized breast epithelial MCF-10A cells. The loss of SELENOF led to increased cell proliferation and abrogated cell death, both of which are hallmarks of carcinogenic cellular transformation. Overexpression of SELENOF in cancer cells with low endogenous SELENOF levels induced cell death via apoptosis and autophagy. These findings collectively provide evidence that loss of SELENOF observed in breast tumors contributes to breast tumorigenesis.

## Figures and Tables

**Figure 1 cancers-15-03671-f001:**
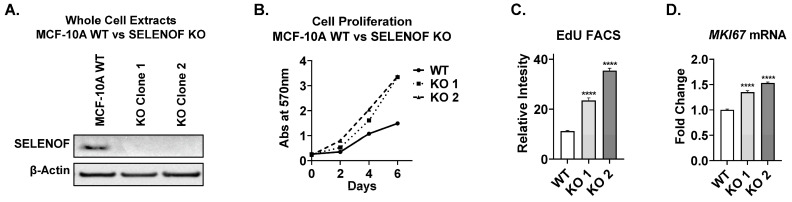
Loss of SELENOF increases proliferation of MCF-10A cells in adherent culture. (**A**) SELENOF levels in whole-cell extracts were determined through Western blotting of MCF-10A wild-type (WT) or single-cell-derived lines from SELENOF knockout cells (KO clones 1 and 2). β-Actin is shown as a loading control. The uncropped blots are shown in Appendix A. (**B**) Cell proliferation was measured over 6 days in culture. Cells were fixed, stained with crystal violet, solubilized in 1% SDS, and quantified by absorbance at 570 nm. (**C**) Relative intensity of EdU-488-stained cells 48 h post-seeding was analyzed using FACS. (**D**) mRNA expression of *Ki67* was examined using RT-QPCR. **** *p* < 0.0001.

**Figure 2 cancers-15-03671-f002:**
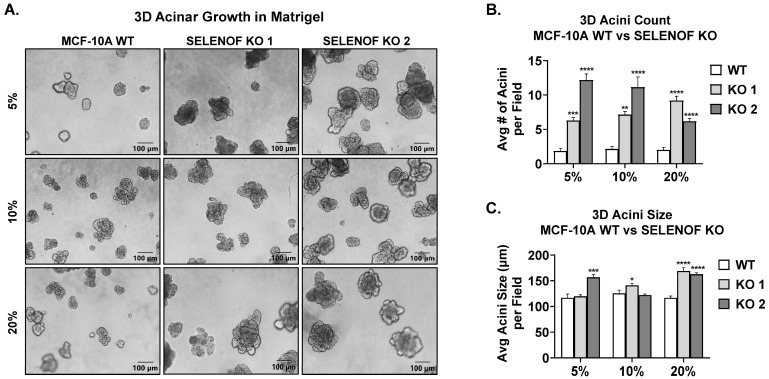
Loss of SELENOF increases three-dimensional acinar growth in the extracellular matrix. (**A**) Representative pictures of acini from MCF-10A WT (first column) or SELENOF KO lines (second and third columns) grown in 5%, 10%, or 20% matrigel for 20 days. Scale bar indicates 100 µm. (**B**) Average number of acini (>100 µm in diameter) per field (n = 3 wells, 7 fields each) plotted for each cell line. (**C**) Average acini diameter size per field (n = 3 wells, 7 fields each) calculated for each cell line. * *p* < 0.05, ** *p* < 0.01, *** *p* < 0.001, **** *p* < 0.0001.

**Figure 3 cancers-15-03671-f003:**
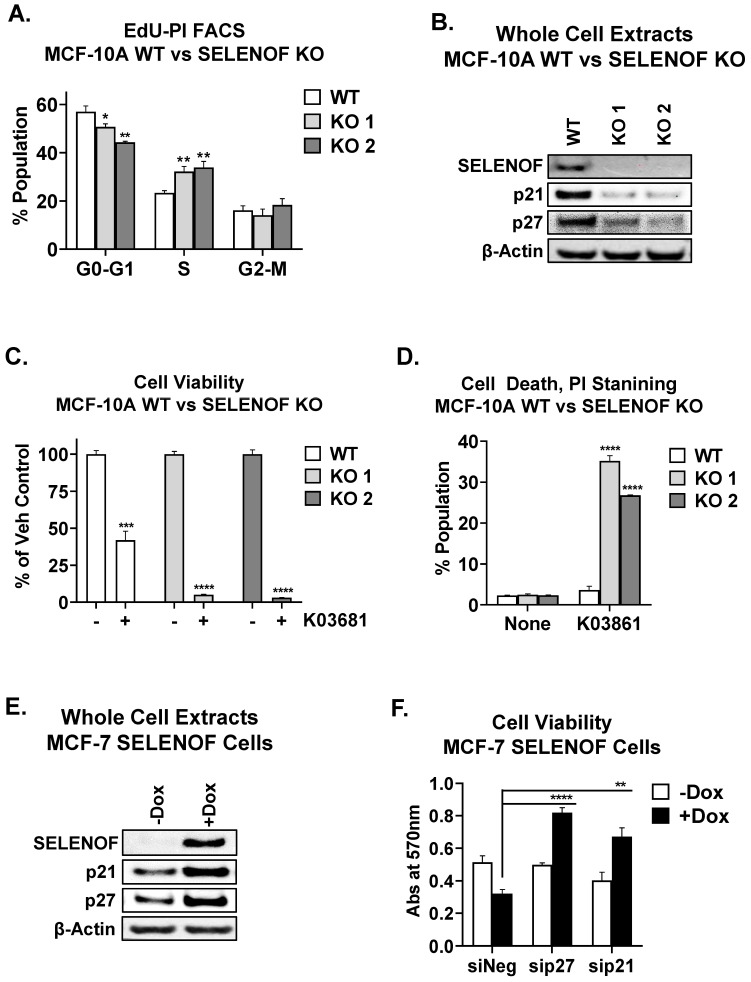
SELENOF controls proliferation and cell death via p21 and p27. (**A**) Cell cycle distribution was determined in MCF-10A WT and SELENOF KO cells stained with EdU and PI and analyzed using FACS. (**B**) The levels of SELENOF, p21, and p27 were determined using Western blotting. β-Actin is shown as a loading control. (**C**) Cell viability of MCF-10A WT vs. SELENOF KO cells was determined through crystal violet staining after 72 h of treatment with −/+ 2 µM K03681. Data were normalized to vehicle controls and shown as 100% for each cell line. (**D**) Cell death was estimated using PI staining and FACS analysis in cells treated for 48 h with −/+ 2 µM K03681. (**E**) Levels of SELENOF, p21, and p27 were determined through Western blotting. β-Actin is shown as a loading control. MCF-7 SELENOF cells were treated with −/+Dox 1 µg/mL for 3 days. The uncropped blots are shown in Appendix A. (**F**) Cell viability was determined using crystal violet staining. MCF-7 SELENOF cells were transfected with siNeg, sip21, or sip27, 10 nM each, for 48 h, and treated with −/+ 1 µg/mL Dox for an additional 72 h. * *p* < 0.05, ** *p* < 0.01, *** *p* < 0.001, **** *p* < 0.0001.

**Figure 4 cancers-15-03671-f004:**
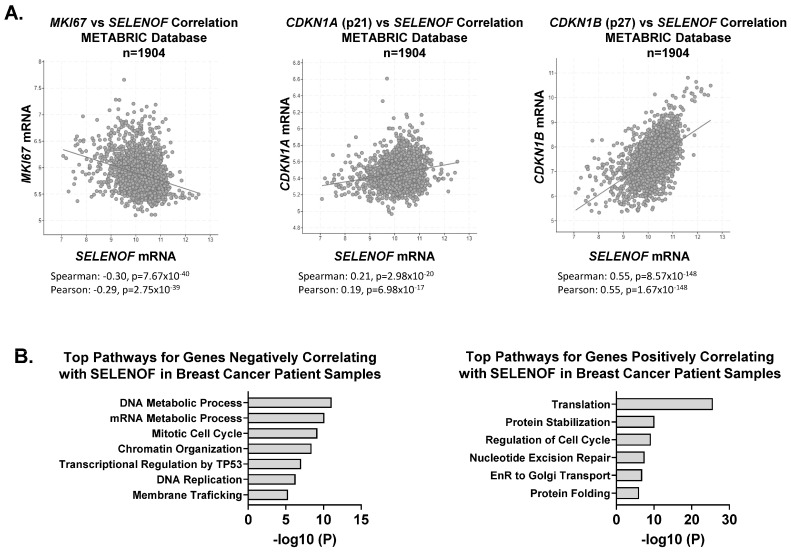
SELENOF expression and its related gene signatures correlate with cell cycle regulators in breast tumors. (**A**) Correlation plots generated to analyze the expression of SELENOF versus Ki67, p21, or p27 in breast tumors from the METABRIC database, n = 1904 samples. Spearman and Pearson coefficients and their respective *p*-values are indicated. (**B**) The top 400 genes co-expressed (negative correlation in the left panel versus positive correlation in the right panel) with SELENOF in breast tumors were analyzed for pathway enrichment using the Metascape gene annotation software (Metascape.org). The −log of *p*-values are shown in the graphs.

**Figure 5 cancers-15-03671-f005:**
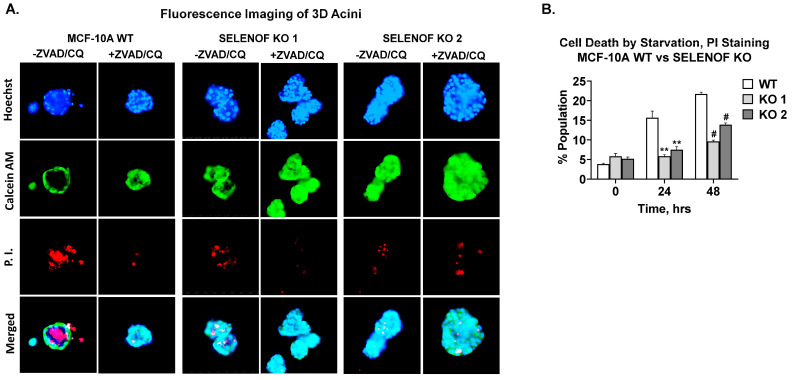
Loss of SELENOF renders MCF-10A cells resistant to apoptosis and autophagy. (**A**) Fluorescence images of 3D acini at 4× generated from MCF-10A WT or SELENOF KO cells. All cells were seeded in 5% matrigel for 5 days prior to treatment with −/+ 10 µM ZVAD and −/+ 5 µM CQ, then grown for another 15 days. The first row shows representative images of Hoechst 33,342 staining of nuclei in blue. The second row shows representative images of Calcein AM staining of live cells in green. The third row shows representative images of propidium iodide (PI) staining of dead cell in red. The last row shows a merged image of all three stainings. (**B**) Cell death was estimated using FACS of cells stained with PI after 24 or 48 h of serum deprivation. ** *p* < 0.01, # *p* < 0.0001.

**Figure 6 cancers-15-03671-f006:**
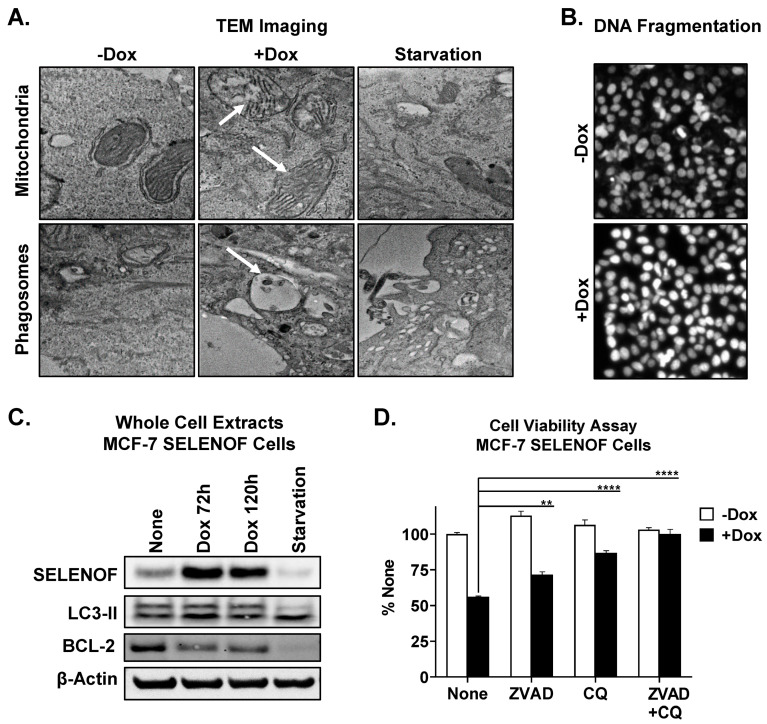
SELENOF overexpression induces apoptosis and autophagy in breast cancer cells. (**A**) Transmission electron microscopy analysis of MCF-7 SELENOF cell treated with −/+ 1 µg/mL Dox for 72 h. Examples of the observed mitochondrial morphology (top row) and phagosome formation (bottom row) are indicated by arrows. Serum starvation (last column) for 24 h was used as a positive control for autophagy. (**B**) DNA fragmentation was estimated using Hoechst 33,342 staining in cells treated as described in (**A**). (**C**) The levels of SELENOF and LC3-II were determined using Western blotting in whole cell extracts from MCF-7 SELENOF cells treated with 1 µg/mL Dox at the indicated times. Serum starvation for 24 h was used as a positive control for autophagy. β-Actin is shown as a loading control. The uncropped blots are shown in Appendix A. (**D**) Cell viability was determined using crystal violet staining of cells treated with −/+ 1 µg/mL Dox, −/+10 µM ZVAD, and −/+5 µM chloroquine (CQ) for 5 days. Data are normalized to vehicle control and shown as 100%. ** *p* < 0.01, **** *p* < 0.0001.

## Data Availability

The data can be shared upon request.

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
