# Peer review of "SELENOF Controls Proliferation and Cell Death in Breast-Derived Immortalized and Cancer Cells"

_cancers, 2023, doi:10.3390/cancers15143671_

Round 1

Reviewer 1 Report (Previous Reviewer 1)

The authors have addressed all the previous concerns thoroughly and hence, the manuscript is now deemed suitable for publication.

Reviewer 2 Report (Previous Reviewer 2)

Authors have improved their manuscript. I do not have other comments.

This manuscript is a resubmission of an earlier submission. The following is a list of the peer review reports and author responses from that submission.

Round 1

Reviewer 1 Report

The manuscript by Ekyalongo et al investigated the specific role of SELENOF loss in immortalized breast epithelial MCF-10A cells and the effect of SELENOF overexpression in MCF7 breast cancer cells. Previous studies from the same group indicated SELENOF is downregulated in breast cancer cells and restoring SELENOF functions can attenuate breast cancer cell growth both in vitro and in vivo. In the present study, the authors demonstrated that knocking down or overexpressing the protein has opposing effects, as indicated in cell proliferation, 3D acini formation, EM studies, and cell cycle proteins p21 and p27 regulate the SELENOF-mediated anti-tumor effects. The results are significant and will contribute to our present understanding of the function of SELENOF in breast cancer etiology. However, there are some issues/concerns which need to be addressed before consideration for publication.

Fig 3: (i) Authors claimed that SELENOF deletion reduced cell proliferation by inhibiting p21/p27 proteins which activate pro-proliferative CDK2, but no data was shown regarding CDK2 levels (mRNA or protein) in SELENOF KO cells or with KO3861 treatments to establish the specificity or minimum off-target effects of the experimental design. Even with the p21 or p27- siRNA transductions (Fig 3F), CDK2 levels were not checked and should be included in the main figure or supplemental data.  

(ii) Fig3C showed reduced cell viability with K03681 but Fig 3D showed no significant increase in cell death in the MCF-10A cells. Does that indicate a cell senescence induction with CDK2 inhibition or any other forms of growth inhibitory pathways in these cells?

(iii) In Fig3E, SELENOF induction has induced p21/p27 levels, but additional experiments should be done to nullify the effect of doxycycline alone on p21 and p27 levels (eg. treating the TetON plasmid transfected cells with dox)

Figure 5 is mentioned wrongly as Figure 6 throughout the result section, leading to confusion, this error must be corrected before resubmitting the paper.

In Figure 6A (or Fig 5A?), acini formed by SELENOF KO cells have a high Calcein-AM/PI ratio indicating their ability to resist apoptotic/autophagic cell death (3rd and 5th columns). In this context, the purpose of using ZVAD/CQ is not clearly understood. Instead, better control would have been the use of an apoptosis/autophagy inducer. A similar experiment with MCF7 cells (+/- Dox induction) can be included to show the SELENOF-mediated inhibition of lumen formation, substantiated with apoptosis/autophagy marker western blot.

Fig S4 should be supplemented with a corresponding western blot to show the changes in the respective protein levels.

The authors should include justifications for all the dosages of the inhibitors used in this study. How are the different concentrations optimized? 

It is important to add a discussion about the significance of the SELENOF-CDK2 signaling axis in breast cancers. The data presented in Fig3 showed that loss of SELENOF is sensitizing the cells to CDK2 inhibition, a mechanism that is now gaining importance in breast cancer therapeutics, especially in cancers with low SELENOF expression.

Reviewer 2 Report

Thanks for inviting me to review. The data can confirm the credibility of the results. I noticed that reviewer 1’s comments on the results are professional and authors can improve the quality of the results based on those comments. However, authors should add a section including “strength and limitation” in the discussion. For example, the significance of this study is limited, as the mechanisms it presents are rather one-sided. Only p21 and p27 were tested, and the positive results led to the conclusion that SELENOF exerts its effects through these two genes. However, it is possible that SELENOF affects a multitude of downstream pathways, which cannot be determined by Western blot analysis alone, and would require sequencing to investigate. Thus, it may be necessary to add a “potential” in “largely via p21 and p27”. Furthermore, the statement that "SELENOF is a breast tumor suppressor" is already established knowledge, therefore, the study fails to offer very novel insights, whereas it may be a good addition to the previous viewpoint. Additionally, the implication of the study should be mentioned, such as the significance of the finding based on SELENOF.

Reviewer 3 Report

The current article by Ekyalongo is an interesting article that briefly discusses about the role of SELENOF in controlling proliferation and death of breast cancer cells. The article is interesting, but it has several issues.

The authors have not provided any of the figures supporting their data? Please provide them in the revised version.

Some of the sentences seems to be too lengthy that makes their meaning unclear. Authors can either reframe them or split them in two for a better understanding.

Please check the font type and size for the references. It is different from main text.

Out of 30 references only one or two are from 2023. Please discuss the paper in the light of recent and latest literature available

The article lacks conclusion section. Please add that to the script.

The authors have proved the role of SELENOF in cell death and controlling cell proliferation. How do the authors’ see its role in developing some novel therapeutics? What can be the other prospects? Please discuss.

Some minor issues:

Line 39 to 40: The meaning of sentence is not clear.

Line 227: Please check the English of the sentence.

Line 254 to 258: These are redundant with the Results section.

English language is fine.
